# Understanding the links between human health, ecosystem health, and food systems in Small Island Developing States using stakeholder-informed causal loop diagrams

Leonor Guariguata[1,2], Gordon M. Hickey[1], Madhuvanti M. Murphy[2], Cornelia Guell[3], Viliamu Iese[4], Karyn Morrissey[5], Predner Duvivier[6], Stina Herberg[7], Sashi Kiran[8], Nigel Unwin[3,9] *

**1** Department of Natural Resource Sciences, Faculty of Agricultural and Environmental Sciences, McGill University, Ste-Anne-de-Bellevue, QC, Canada, **2** George Alleyne Chronic Disease Research Centre, University of the West Indies, Bridgetown, Barbados, **3** European Centre for Environment and Human Health, University of Exeter, Exeter, United Kingdom, **4** Pacific Centre for Environment and Sustainable Development, University of the South Pacific, Suva, Fiji, **5** Department of Technology, Management and Economics, Technical University of Denmark, Lyngby, Denmark, **6** Faculté d'Agronomie et de Médecine Vétérinaire, Université de d'État d'Haïti Port-au-Prince, Port-au-Prince, Haiti, **7** Richmond Vale Academy, Richmond, St Vincent and The Grenadines, **8** Foundation for Rural Integrated Enterprises and Development (FRIEND), Lautoka, Fiji, **9** MRC Epidemiology Unit, University of Cambridge, Cambridge, United Kingdom

* nigel.unwin@mrc-epid.cam.ac.uk

## Abstract

Globalized food systems are a major driver of climate change, biodiversity loss, environmental degradation, and the increasing prevalence of overweight and obesity in society. Small Island Developing States (SIDS) are particularly sensitive to the negative effects of rapid environmental change, with many also exhibiting a heavy reliance on food imports and high burdens of nutrition-related disease, resulting in calls to (re)localize their food systems. Such a transition represents a complex challenge, with adaptation interventions in one part of the food system contingent on the success of interventions in other parts. To help address this challenge, we used group model-building techniques from the science of system dynamics to engage food system stakeholders in Caribbean and Pacific SIDS. Our aim was to understand the drivers of unhealthy and unsustainable food systems in SIDS, and the potential role that increased local food production could play in transformative adaptation. We present two causal loop diagrams (CLDs) considered helpful in designing resilience-enhancing interventions in local food systems. These CLDs represent 'dynamic hypotheses' and provide starting points that can be adapted to local contexts for identifying food system factors, understanding the interactions between them, and co-creating and implementing adaptation interventions, particularly in SIDS. The results can help guide understanding of complexity, assist in the co-creation of interventions, and reduce the risk of maladaptive consequences.

**Data Availability Statement:** All relevant data are within the paper and its Supporting Information files.

**Funding:** Much of the work presented in this paper has been undertaken as part of two projects funded by UKRI GCRF(BB/T008857/1 to NU, MM, GH, VI, CG, KM, PD and MR/PO25250/1 to NU, CG, VI) and we gratefully acknowledge their funding support. This research has also received funding through the University of Exeter's UKRI Global Challenges Research Fund (GCRF) quality- related research fund (KM, CG). The authors and the work received no commercial funding. The funders had no role in study design, data collection and analysis, decision to publish, or preparation of the manuscript.

**Competing interests:** The authors have declared that no competing interests exist.

## Introduction

The complex inter-relationships between human health, ecosystem health and food systems sustainability have been gaining global policy attention, as highlighted by the inaugural United Nations Food Systems Summit held in 2021 which sought "to transform the way the world produces, consumes and thinks about foods within the context of the 2030 Agenda for Sustainable Development and to meet the challenges of poverty, food security, malnutrition, population growth, climate change, and natural resource degradation" [1]. Globally, food systems contribute to between a quarter and a third of all anthropogenic greenhouse gas emissions and are the major cause of biodiversity loss [2].There is an urgent need to transform the way in which humans use natural resources for food production away from degenerative cycles that damage ecosystems and human health and towards restorative practices that secure food and nutrition and mitigate the negative impacts of climate change [3]. Small Island Developing States (SIDS) have been at the forefront of recent calls for food system transformation due to their unique vulnerability to the effects of climate change including increasing intensity of climatic extremes, sea-level rise and changing environments, and many share a high burden of noncommunicable diseases (NCD) and obesity [4] despite contributing less than one percent of global greenhouse gas emissions [5].

One of the main contributors to the high burdens of NCDs in SIDS is unhealthy diets that are often high in ultra-processed foods and low in fruit and vegetable consumption [6, 7]. Due to limited land availability and underdeveloped domestic food production systems, SIDS are among the most food import-dependent countries [8]. Food imports in the SIDS have risen from 45% in the early 1990s to 60% by 2011 in the Pacific and over two thirds in the Caribbean [9]. Adapting food systems to promote food security and sovereignty, while reducing vulnerability to extreme weather events and other natural disasters has been a major policy objective in SIDS [10]. The Caribbean Community (CARICOM), for example, has set a commitment to reduce the regional food import bill by 25% by 2025 [11]. However, evidence to date suggests that a high and increasing reliance on food imports continues, alongside increasing burdens of nutrition-related ill health, including obesity and type 2 diabetes.

Food systems are complex socio-ecological systems, with food insecurity reinforcing food system vulnerability [12]. Interventions designed to adapt SIDS food systems towards food security have a long history, and include school feeding programs [13–15], nutrition education programs [16, 17], agricultural education and extension services [18], capacity building [19], community gardening initiatives [20], the formation of agricultural co-operatives [21, 22], the development of improved crop and livestock varieties [e.g., Samoa Agriculture Competitiveness Enhancement Project (SACEP) [23], the Central American and Caribbean Crop Improvement Alliance (CACCIA) [24], and the Caribbean Agricultural Research & Development Institute (CARDI)] [25], investments in storage, transport and processing infrastructures (e.g., the Barbados/ Guyana Food Terminal [26]), the development of water conservation [27] and low input food production techniques including agroecological approaches [28, 29], amongst many others.

The cascading impacts of, and vulnerability to, climate change further complicates these food system interventions that seek to lessen the burden of malnutrition and further environmental degradation. Mitigation of climate change is clearly outside the control of SIDS, given their very low contribution to global greenhouse gas emissions. Adaptation to the effects of climate change is where their focus must lie, with challenges from extreme weather events, rising sea-levels, storm surges, coastal erosion and soil salinization, and declining health of coral reefs and fish stocks. A major challenge undermining the success of food system interventions under climate change is that the "causal vulnerability of food systems is contextual and driven

by multidimensional stressors and actors interacting at many temporal and spatial scales, undermining the capacity of societies to confront multiple-exposures to multiple threats" [12]. Cross-sectoral, cross-spatial and holistic intervention strategies are therefore essential. Further, as noted by Reay et al., [30] food system vulnerability to climate change is inherently local, and any intervention that fails to account for local context risks unintended consequences, including maladaptation (see also Zavaleta et al, [31]). Maladaptation can be defined as a process that exacerbates the existing causes of vulnerability or creates new ones, essentially worsening the situation for society and the environment [32]. Recent examples from SIDS regarding food systems have included efforts to increase crop yields through agricultural intensification practices in São Tomé and Príncipe, which benefited farmers who owned land, but increased the vulnerability of farmers who did not, contributing to inequality and marginalization [33, 34]. Widespread adoption of such practices can also intensify erosion and runoff, increase water pollution, and negatively affect biodiversity and ecosystem services [35].

Maladaptation is particularly important to identify because it is an unexpected and unwanted outcome of a strategy that has been implemented with good intentions [34]. Within SIDS food systems, maladaptation can arise from not being aware of what's driving food insecurity and the negative impacts of environmental change (causal pathways), emphasizing the wrong actors, and not understanding the wider context [34]. Systemic issues of marginalization, exclusion and lack of involvement are known to drive maladaptive trajectories in local food systems [12, 31, 35, 36]. Despite the need to better incorporate complexity science in adaptation intervention strategies, frameworks that link and seek to understand causal vulnerability and food systems to improve resilience remain rare [12]. Such frameworks can help to identify the potential for maladaptation before it happens by soliciting multi-stakeholder input from a diversity of actors working to address human and ecosystem health through local food system change in any particular context. The pressing questions facing these actors are: how best to intervene across the different levels and sectors affecting SIDS food systems to enhance their resilience and sustain positive impact? How to measure intervention success and attribute causation? And once success is identified, how to share and significantly scale up approaches through existing and novel social mechanisms? Answering these questions necessitates research frameworks that embrace complexity, incorporate context and connect outcomes with processes and meanings [12, 37]. The objective is therefore to engage food system actors, including policy makers, in deliberative and reflexive learning processes to better understand ongoing food system challenges in SIDS, identify how to adapt local food systems through community-based intervention packages, and how to monitor and evaluate outcomes. We present two frameworks designed using participatory complex systems thinking methods [38–40] to help identify adaptation trajectories through local food system interventions, drawing on our collaborative experience working with stakeholders along local food value chains in Caribbean and Pacific SIDS.

## Methods

### Study setting

The Intervention Co-creation to Improve Community-based Food Production and Household Nutrition in Small Island Developing States (ICoFaN) [41] project was developed in partnership with local non-governmental organizations (NGOs) developing community-based interventions supporting local food production and consumption in Fiji and St Vincent and the Grenadines (SVG) from February 2020 to May 2023. It also piloted data collection on household nutrition and food production in the Nippes region of Haiti. The project engaged with stakeholders in each of these settings to understand the drivers and barriers to local food

production and consumption. The backgrounds of the SIDS based stakeholders included agronomy (2), disaster risk management and food systems (2), community development (with a focus on food production)(7), and the food systems as determinants of non-communicable diseases (3). Researchers from the UK (3) and Canada (2) had backgrounds in studying food systems, systems thinking and public health impacts of non-communicable diseases.

The international project faced many challenges with the arrival of the SARS-Cov 2 pandemic. For Fiji, strict restrictions meant that we had limited contact with the communities where the project was to be implemented. Further, during intervention implementation in SVG, a catastrophic eruption of the La Soufriere volcano occurred, severely limiting the development of community gardens due to ash falls and the destruction of villages. Over this same time period Haiti faced major political and social disruption, including the assassination of its President in 2021. Our collective experience adapting to these and other shocks informs our findings.

## Participatory group model-building

We conducted two group model-building sessions each with project implementation partner for the study sites separately, and held two additional workshops with all study partners to elaborate causal loop diagrams for understanding the drivers of food insecurity across the study sites, and develop a shared view of how to co-create interventions for community based local food production. The sessions were facilitated by LG, NU, and GH and were conducted virtually due to the SARS-Cov 2 pandemic. We have published elsewhere details of our methods for engaging with stakeholders in the construction of food system causal loop diagrams, which are from those developed and applied in the field of system dynamics [38, 39]. The participants included project implementing partners from two local, well-established NGOs with experience in developing community gardens: the Foundation for Rural Integration and Enterprise Development (FRIEND) in Fiji and Richmond Vale Academy in SVG. Disaster risk management and food system experts from the University of the South Pacific also participated in the group model-building session for Fiji. For Haiti, causal loop diagrams were developed with two co-investigators from the Faculty of Agriculture and Veterinary Medicine, Université de d'État d'Haïti.

## Causal loop diagrams

One of the primary outputs from group model-building workshops are causal loop diagrams that map the network of determinants and outcomes through a causal system to explain the central problem or reference mode [42]. The reference mode can be considered a single indicator or small set of indicators that changes over time. Elaborating the casual loop diagram seeks to explain that behavior over time and develop dynamic hypotheses for how the variables within the diagram may change with relation to each other [43]. These are shown symbolically as a series of interconnected elements with arrows indicating causal direction. An arrow is positive where the driver and the outcome change in the same direction (i.e., an increase in the driver yields and increase in the outcome) and negative where they change in opposite directions (i.e., an increase in the driver yields a decrease in the outcome). Loops are identified within the causal pathways that can be reinforcing (R) (leading to exponential growth or decay) or balancing (B) (leading to goal-seeking and stabilizing behavior). The causal loop diagram is a qualitative exercise and does not include effect sizes, but can serve as a useful tool for discussion and a unified vision of the complex dynamics within a system [44]. For this project, the central reference mode was a combination of rising food-driven poor health (in particular NCDs and obesity), decreasing ecological health (including soil quality, coral reefs) and an

increasing dependence on food imports. Each of these three factors align with the dimensions of the expanded definition of food security proposed by Clapp and colleagues [45], touching on access to and utilization of adequate nutritious food, sustainable (ecologically and socially) food production and the agency of populations over their food systems.

### Incorporating past experience—Framework development

In addition to the causal loop diagrams developed with ICoFaN implementing partners, we also drew from experience and previous participatory group model-building sessions with other related projects [38, 46, 47] in SIDS as well as qualitative research conducted with community members that informed the design of this project [46]. The combined projects have engaged stakeholders along the food value chain to better understand the local contextual factors shaping food sources, availability and utilization. An overarching aim of this engagement was to understand the drivers, rather than just the outcomes, of increasing exposure to unhealthy food systems. Two resulting causal loop diagrams are presented. One serves to understand the system that drives and determines human and ecological health in the SIDS. The second is a framework for implementing co-created community food interventions with a focus on the sustainability of outcomes and improving adaptive capacity.

The ICoFaN project was particularly interested in promoting 'agroecology', and 'agroecological food production' to reduce food system vulnerability and promote sustainability. A core definition of agroecology is 'the science of applying ecological concepts and principles to the design and management of sustainable food systems' [48]. In addition, our use of the term embraces the ten elements of agroecology as described by the UN Food and Agriculture Organization [49], which emphasizes the need to consider the social, cultural and economic aspects of sustainability within food systems [50]. We acknowledge that other terms are also widely used to denote sustainable approaches to food production, such as 'regenerative agriculture'. Our preference for 'agroecology' is supported by a recent study examining the use of different terms, which concluded that, "agroecology has progressed the furthest in defining pathways towards food system transformation, guided by principles that have been widely legitimized in local and global policy spaces [51]".

## Results

### Causal loop diagram 1—The system determining human and ecological health in food production in the SIDS

In partnership with stakeholders and drawing from past experience, Fig 1 illustrates a causal loop diagram that shows the relationships between the drivers, determinants, and outcomes of the food system and its links to human and ecological health in the SIDS. Some key pathways and feedback loops describe the complex situation facing SIDS and enable the formation of dynamic hypotheses on how interventions may be affected by or change that situation.

We identify four major categories of actors in the generalized system determining food production in the SIDS: political actors (e.g. elected official and decision makers), conventional commercial interests, agroecological food interests and communities. The actions of each of these actors are interlinked to form the system that determines the type of food production, its outputs, and its effects on human and ecological health. In what follows we examine each of these actors in turn and their links in the system through the feedback loops described in Fig 1. These are labeled as either R for reinforcing, which leads to exponential growth or decay of the variables connected to them, or B for balancing, which leads to a goal-seeking and stabilizing effect that tends to dampen the effects of reinforcing loops.

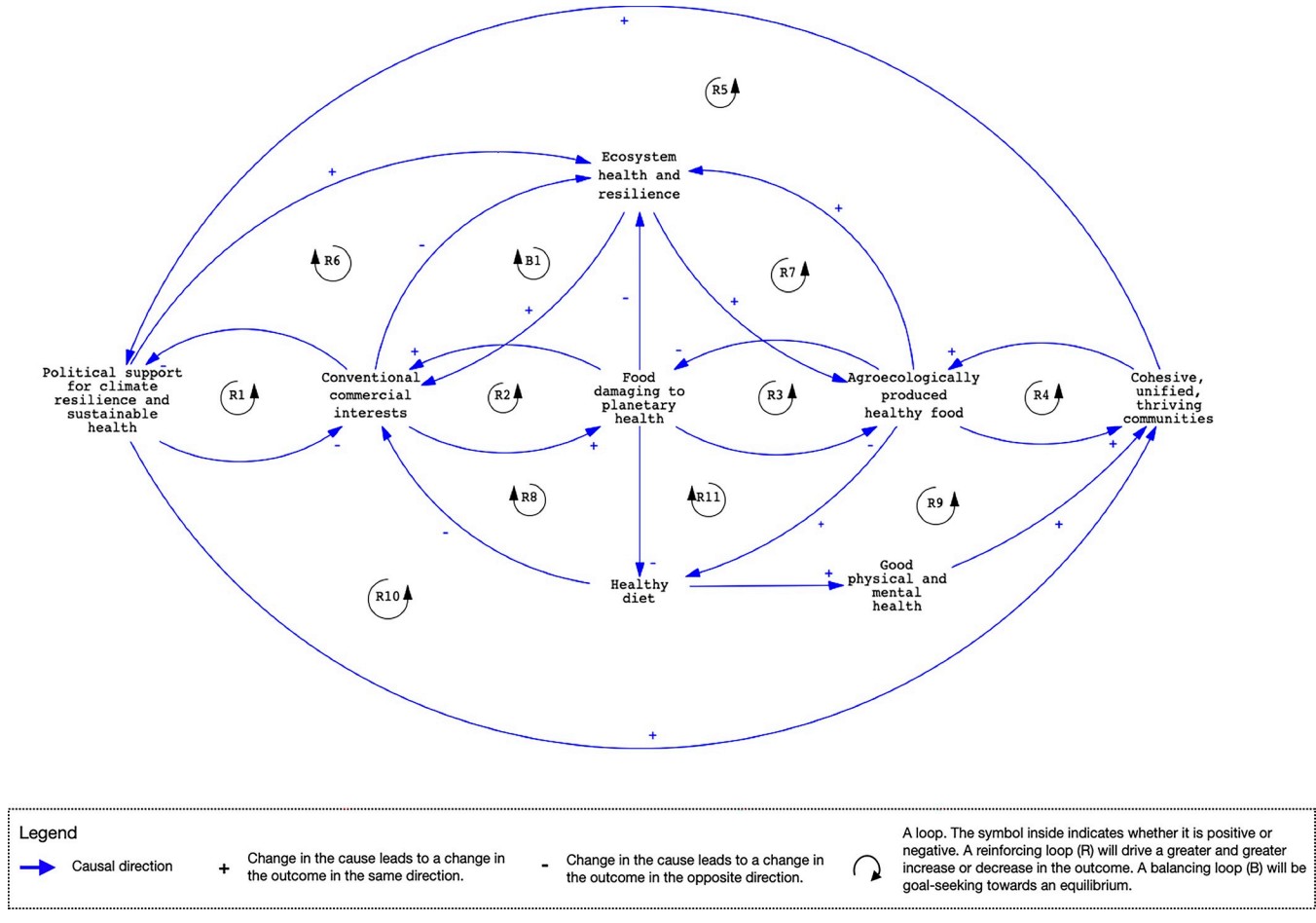

**Fig 1. A causal loop diagram of the system determining human and ecosystem health in SIDS food systems.** Causal pathways are identified as reinforcing (R) (leading to exponential growth or decay) or balancing (B) (leading to goal-seeking and stabilizing behavior). Each of the identified pathways are summarized and further described in S1 Table.

## Political determinants of sustainable health and climate resilience

Stakeholders identified that the policy actions and support of government have a significant role on the health of communities and environments. In the SIDS, this role may be limited at the international level where they may not have the political capital necessary to control agreements that ultimately affect the food system [52]. However, local action and regulations can have an impact. In the SIDS, political support currently reinforces commercial interests, those interests have an influence on politics both through economic means and also directly via trade agreements, lobbies, and contributions as described in R1. Political actors may not want to take the risk of alienating these interests and losing their political capital and this undermines climate resilient policy development. Stakeholders in Fiji described how aid packages, increasingly necessary in light of more frequent and more intense storm seasons, often contain a high proportion of imported ultra-processed foods on which the population becomes dependent in the aftermath of cyclones. These packages are provided by foreign donors with little regulation from local governments, making them beholden to the donors regardless of the health and social impacts on communities.

A climate supportive political environment that limits the effects of harmful agricultural practices, degrading ecological development, and strengthens climate resilient strategies in

communities could reduce the influence of commercial interests, at least at the local level. Political support for climate resilience would also strengthen policies for ecosystem health which in the long term could sustain healthy food production, healthy diets, and reduce the influence of conventional commercial interests (R6). Stakeholders identified this support as being undermined by an outsized influence of conventional commercial interests which often drives political agendas through direct contributions or through economic influence as employers or investors in SIDS.

## Conventional commercial interest and ecological destruction

Our workshops also identified that conventional commercial interests, especially in the food system, contribute to ecosystem-damaging agricultural practices [53], plastic pollution [54] and consumption of ultra-processed foods [55, 56]. These commercial actors can include large multinational corporations that have a large market penetration, but also smaller local producers of unhealthy foods or agricultural producers engaged in conventional farming practices high in fertilizer and pesticide use that degrade land and sea ecosystems [57]. Commercial actors are characterized as a fossil-fuel dependent food supply that is damaging to human and ecological health (R2).

In Fig 1, the negative ecological impact of conventional practices leads to lower agricultural production due to soil degradation and depletion, creating a greater need for chemical fertilizers (B1). These fertilizers runoff into the water ways and oceans depleting reefs and fishing stock further degrading the food supply and fishing capacity [58], creating a greater dependence on imported foods for communities which are often perceived as safer even though there may also be pesticide residues present (R8). The production of ecosystem-damaging foods can also undermine 'healthy diets' through the presence of pesticides in fresh agricultural products (which can also damage the health of farmers handling the pesticides and consumers). Due to perceived pesticide contamination of locally produced foods, stakeholders in workshops in the Caribbean described a lack of confidence in the quality of local agricultural products among the community, preferring to consume imported foods, which it is assumed to come from settings with better regulation and thus where residual pesticide contamination is less likely [38]. Consumption of these foods increases the revenue of the commercial producers, enabling them to continue marketing and selling their products to the community (R8) or for export [59] and, in the absence of regulation on food quality standards or chemical fertilizer and pesticide use, strengthening their political influence (R1).

## Agroecological food interests

Agroecology offers potential solutions to the challenges associated with designing more sustainable and resilient food production systems in ways that engage with and build on local knowledge and food cultures [49]. However, according to our participants, in the SIDS, there are no large-scale agroecological food interests in the way that conventional commercial actors exist. This is, in part, because agroecological practices aim to benefit local communities and the environment more than maximizing profits [60]. However, we envision the role that a movement centered on agroecological food interests could have in the food system, recognizing that at present the role is small. Agroecological food production practices generally compete with conventional monocrop food production practices in SIDS, particularly in terms of farmer time and interest, access to local markets, available arable land, and water resources (R3). However, over time, agroecologically produced foods can help to restore local ecosystems and contribute towards the adaptive resilience of the food system through more sustainable food production in a reinforcing loop (R7). Agroecological approaches to improving and

maintaining soil health, the use of integrated pest management techniques, reducing the need for chemical pesticides, and synergies gained through locally appropriate mixed cropping systems are some examples of how this might be achieved. The production of agroecologically produced healthy foods can also reinforce cohesive and thriving local communities when driven by community-based principles including local ownership and agency (R4 and see Fig 2) and reinforce better health outcomes driven by healthy diets and the practice of

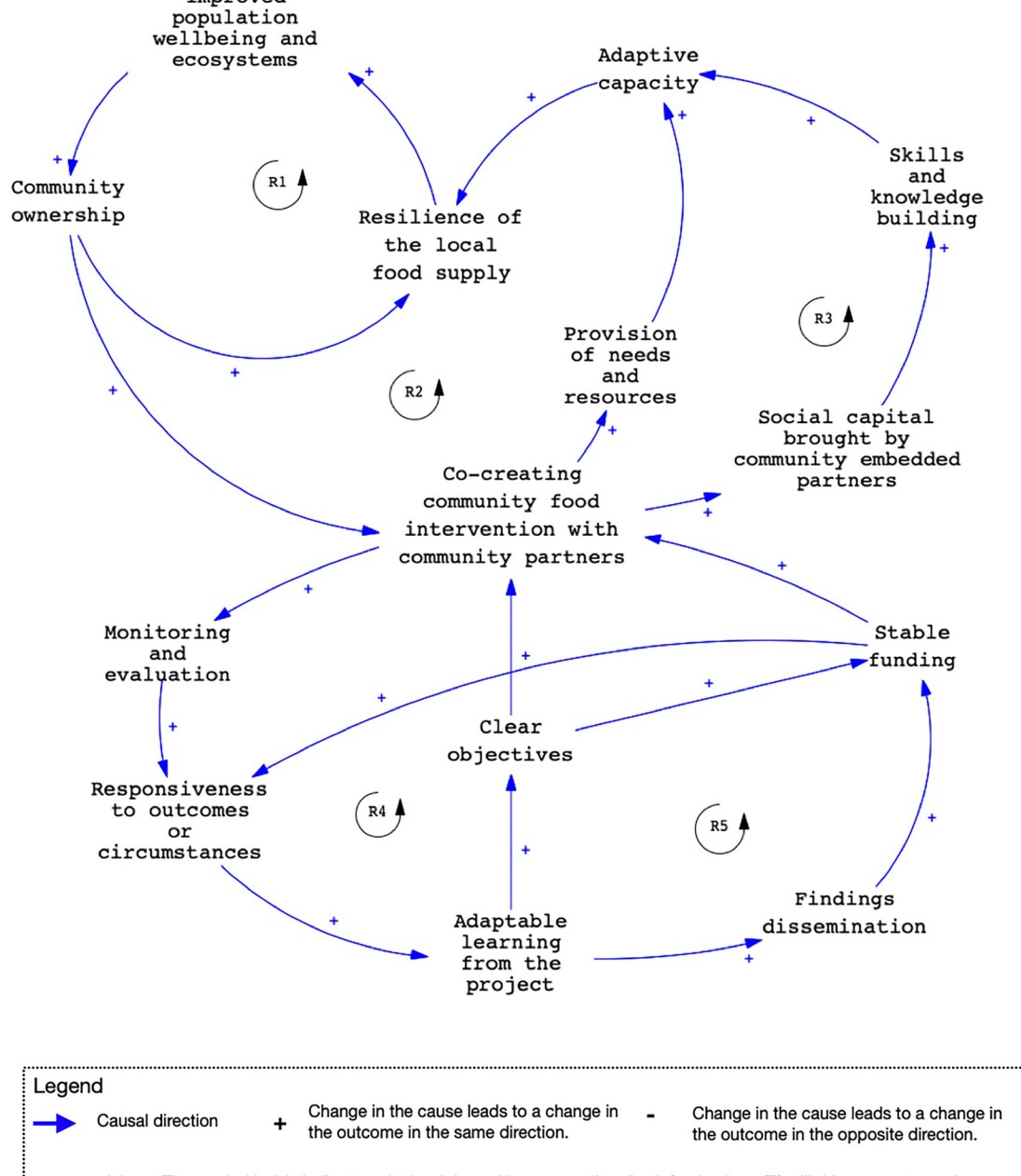

**Fig 2. A framework for co-creating community food intervention with community partners.**

community-based farming (including physical and mental health) further creating a demand for agroecological food production (R9).

## Community members

The cohesiveness of communities can have a strong influence on their capacity to adapt to change, mitigate problems, and improve well-being [61]. However, community cohesion is a multi-dimensional phenomenon that can vary considerably over time depending on the issues at stake. In the context of transitioning to more localized food systems, cultivating and sustaining the necessary levels of community cohesion presents a major challenge that requires ongoing processes of dialogue, frequent interaction, and participation to overcome [62]. In our framework, we consider just one approach that involves engagement of local communities in food production. Overall, maintaining healthy diets and the attendant physical and mental health benefits could strengthen cohesive communities and potentially build their capacity to produce food in community gardens, reducing the use of ecosystem damaging foods and reinforcing health (R9). These benefits could also help to reduce the influence of conventional commercial interests as fewer people consume their products. The relationship with adaptive, cohesive communities to agroecological farming is described in more detail in Fig 2. Cohesive communities can have an influence on local governments and increase demand for climate resilience and sustainable health policy (R5) which can then further strengthen the health and well-being of those communities. Such community-driven political support could potentially increase agroecological food production, reduce the influence and reach of ecosystem damaging foods and their producers, further strengthening political support and collective work to restore ecosystems (R10).

## Causal loop diagram 2—A framework for developing and assessing co-created community food interventions in the SIDS

Interventions for developing ecologically sustainable or restorative community farming in SIDS need to be adaptive to the evolving realities of the community; engage with community-embedded partners; use a co-creation participatory approach; and have as a key objective the strengthening of the capacity of the community to adapt to external shocks (vulnerability reduction). While Fig 1 describes a generalized food production system and its consequences in the SIDS, Fig 2 describes the necessary components and assumptions underlying co-created community food system adaptation interventions based on our experience gained over multiple projects working in partnership with key stakeholders in SIDS. Table 1 first defines the major actors and their roles in the intervention process presented in Fig 2. Each of the pathways identified in Fig 2 are summarized and further described in S2 Table.

The vision for implementing co-created community food system interventions by our stakeholders can be described by the relationships presented in Fig 2 (R1). Building the resilience of local food supply chains will add to the well-being of the local population and support the health of ecosystems. This in turn should strengthen community ownership and participation in the production of resilience-enhancing local foods. Key to the success of this vision is the adoption of agroecological principles such as diversity, synergy, efficiency, resilience, recycling, co-creation and sharing of knowledge [50] to protect against the degradation of ecosystems and the resulting negative effects on planetary health.

Realizing such a vision also relies on the intervention being successful enough to provide the necessary impulse to set off a reinforcing loop R2 that can become self-sustaining in support of social learning and adaptive capacity, contributing to the loop in R1. Greater community ownership should improve processes of co-creation, engagement, and sustainable uptake

**Table 1. Roles of key actors in the co-creation of food system interventions by communities and researchers.**

| Actor | Role | Description |
|---|---|---|
| Community members | The primary beneficiaries of the intervention consisting of households living within a defined foodshed. Ideally, co-creators, implementers and evaluators of the intervention. | • Participants should be identified based on the potential for them to experience sustained benefits and their ability and willingness to engage with the intervention. |
| Community-embedded partners | On-the-ground experts in the development and implementation of local ecologically sustainable food production. Provide a guiding role in the community passing on essential knowledge and skills, ensuring the provision of project inputs, and engaging with the key community stakeholders to ensure buy-in. Provide a key avenue of communication with researchers and participate in implementation co-creation. | • The partner should have a longstanding, successful relationship with the target community.<br>• They should be accepted and recognized by community members as partners adding value and engaged with the needs of the community.<br>• They often have a strong relationship with community leaders.<br>• They are experts in the implementation of the co-created intervention and can provide key knowledge to researchers and community-members alike. |
| Researcher | Provides expert knowledge and theory underlying the implementation and evaluation of the study. Secures funding for research and disseminates wider results to the scientific community and to the target communities. | • Brings scientific expertise and rigor to the development, implementation, and evaluation of the intervention.<br>• Has a vested role in improving outcomes for the community.<br>• Secures funding and communicates findings and objectives adapting to the needs of the community.<br>• Evaluates findings to contribute to scientific knowledge and further implementation research. |
| Funder | Provides the resources necessary for the implementation of the intervention. | • Is responsive and adaptive to the needs of the intervention in communication with the researchers, community members, and community-embedded partners.<br>• Understands and values the objectives of the intervention and the research outcomes. |

of local food system interventions. R3 highlights the critical importance of implementing such interventions together with community-embedded partners who bring the necessary social capital to ensure meaningful engagement and facilitate conflict resolution among the stakeholders, decision-makers, and community leaders essential to intervention success.

Based on our experience working in SIDS, the extent to which a co-created intervention will yield gains for the participating community (R1, R2, and R3) in terms of health, adaptive capacity, and ecologically restorative agricultural practices, requires carefully designed monitoring and evaluation (R4) procedures to minimize the risk of inadvertently initiating maladaptive trajectories (e.g., increasing food insecurity and vulnerability to the negative impacts of climate change). R4 connects monitoring and evaluation to learning and adaptation in the design and implementation of co-created community food system interventions to meet the needs of participating communities. It also connects to the need for stable funding (R5) to enable the research team to be responsive to circumstances, facilitate adaptation and support the dissemination of findings to members of the community, key stakeholders, the scientific literature, and the funding agency. Fig 2 shows that disseminating findings can help reinforce stable funding if the research team can demonstrate positive impact from the intervention.

## Discussion

Food systems are major contributors to climate change, are adversely affected by it, and are failing to deliver good nutrition for large sections of humanity [63]. Planetary health, which must include equitable human nutrition within planetary boundaries, is only achievable with major food system reform [2]. Moreso than in many other parts of the world, this is illustrated by the situation in SIDS, with the need for reform towards food systems that are more self-sufficient and sustainable, able to adapt to the effects of climate change and deliver better nutrition. Given the complex and contextual nature of food systems, there is no one size fits all solution, and guidance on transformation to sustainable food systems emphasizes the

importance engaging and learning from local communities and from stakeholders across the food value chain [49, 64, 65].

In this paper we draw on several years of experience in working with community organizations and other food system actors in the Caribbean and Pacific. We have used participatory systems mapping to derive understandings of local food systems and approaches to intervening in them. We present here two causal loop diagrams that we believe provide useful starting points for others interested in seeking to understand and intervene in SIDS food systems. The causal loop diagram in Fig 1 shows the major food system components and their relationships that determine ecosystem health and human population nutrition. These food system components represent the actions of four main actors: political actors (e.g. elected officials and decision makers), conventional commercial interests, agroecological food interests and communities and their members. We suggest that a good entry point for seeking to understand the complexity, interrelationships and potential leverage points for intervention within a SIDS is to identify these actors and describe their interactions.

Fig 1 illustrates the complex and interconnected nature of local food systems in SIDS and thus the need for multi-level, multi-sectoral and multi-actor coordination to deliver mutually reinforcing interventions [66]. Using Fig 1 as a framework provides an initial dynamic hypothesis for future research that can be modified and elaborated based on locally collected evidence.

Fig 2 identifies the major components of intervention co-creation in SIDS to reduce the risks of maladaptation, paying close attention to learning and inclusion throughout the design, implementation and evaluation of interventions. Fig 2 is generated from our experience on funded research projects in which there is engagement and co-creation between researchers, civil society organizations (including NGOs), government stakeholders, local private sector actors and community members. A key learning point is the importance of working with what we term 'community-embedded partners.' These are partners that can bring invaluable social capital to a project (illustrated in Fig 2) through established relationships with other stakeholders, including members of households in the community, relevant government ministries, local food producers and retailers, and so on. With the community-embedded partners further work can be undertaken as appropriate to observe and learn from the 'autonomous adaptation' occurring within households, rural communities and along local supply chains in response to chronic and acute shocks [67]. This can provide direction for resilience-enhancing intervention design and evaluation for planetary health, with priority given to the most vulnerable.

Building from the framework presented in this paper, members of our team are collaborating on an international research partnership seeking to co-design and evaluate local food system interventions based on agroecological principles for better nutrition and ecological health in SIDS (https://communityfoodplanetaryhealth.org/). Working closely with community-embedded NGO partners in Fiji, St Lucia, St Vincent and the Philippines island of Palawan, the 'Community Food for Planetary Health' project is engaging community members in each setting to develop shared understandings of the context and causes of local food system vulnerability in order to adapt local food system intervention packages and guide data collection, monitoring and evaluation. The outcomes of the interventions, and their scalability, will then be modelled against 'business as usual' to inform national and regional policies.

## Strengths and limitations of causal loop diagrams and group model-building

Group model-building provides a participatory approach to complex thinking that can bring together a wide variety of stakeholders with unique knowledge of a complex problem and

create a consensus framework for hypothesis generation and testing [68]. It is a flexible approach that can be applied in a variety of settings (e.g. face-to-face, in virtual sessions, over several sessions or in one single session) and places control of the causal mapping firmly in the hands of the stakeholders by asking them to describe and draw the casual relationships, describe feedback loops, and identify places where interventions may be most useful [69]. That flexibility allowed us to conduct virtual group model-building sessions when face-to-face meetings were severely limited during the SARS-Cov 2 pandemic. Stakeholders also benefit from engaging in systems thinking, which may allow them to further develop skills for understanding complex problems [70]. However, some limitations to causal loop diagrams and group model-building exist. For example, qualitative representations of the intricate relationships and feedback loops experienced 'on the ground' by stakeholders can be difficult to parameterize within statistical models for further analysis. In that respect, causal loop diagrams should be used primarily for developing dynamic hypotheses to understand the drivers and outcomes in a complex problem, as we have described here, and not to hypothesize on effect sizes for change. In addition, causal loop diagrams are sensitive to the stakeholders present in constructing them and care must be taken to choose from a wide variety of participants across the system. We overcome this limitation, in part, because the diagrams presented here are a summary of a large number of group model building sessions that engaged stakeholders across the food system value chain.

## Conclusion

In this paper we address the need to understand food systems within SIDS as complex socio-ecological systems, and to use this understanding as the basis for the co-creation of coordinated interventions aimed at improving human nutrition, mitigating climate change and building resilience. Based on group model building sessions with stakeholders in SIDS we presented two complementary causal loop diagrams: the first providing an overarching summary of interactions between four main groups of food system actors, and the second providing a framework for the design and implementation of food system interventions. While these diagrams are based on our experience of engaging with food system actors in SIDS over many years, they should be seen as providing dynamic hypotheses to be subject to further evaluation, modification and elaboration. Global food systems are second only to energy production as a source of anthropogenic greenhouse gas emissions, and the ensuing climate change is itself making adequate food production more difficult. Although the contribution to global greenhouse gas emissions from the SIDS is relatively small, a reliance on food imports generated by conventional farming practices and the associated need for shipping links SIDS to this damaging feedback loop. In addition, fossil-fuel driven conventional farming practices within SIDS contribute to the degradation of local ecosystems. SIDS are particularly vulnerable with increasing dependence on food imports, increasing strength of extreme weather events and high burdens of poor nutrition. Transformation towards more sustainable, resilient and healthy food systems is urgently required. We hope that the diagrams we have developed and share here will also be useful to others in that endeavor.

## Supporting information

**S1 Table. Description of the feedback loops in Fig 1 (R = reinforcing; B = balancing).** (DOCX)

**S2 Table. Description of the feedback loops in Fig 2 (R = reinforcing).** (DOCX)

## Acknowledgments

We express thanks to other members of the ICoFaN project team for their support with this review, particularly Dr Robers Pierre Tescar and the students at the Université d'État d'Haïti and Dr. Jioje Fesaitu and colleagues at the University of the Pacific. We would also like to thank our implementing partners at the Foundation for Rural Integrated Enterprises & Development (FRIEND) Fiji and Richmond Vale Academy, St Vincent & the Grenadines for their valuable insights into local food interventions in the Caribbean and Pacific regions. We acknowledge other members of the ICoFaN project team who have contributed to the overall project design and implementation, including: Prof Nita Forouhi, University of Cambridge; Dr Emily Haynes, University of Exeter; Cassandra Halliday, University of Exeter, and Eden August, the University of the West Indies.

## Author Contributions

**Conceptualization:** Leonor Guariguata, Gordon M. Hickey, Madhuvanti M. Murphy, Cornelia Guell, Viliamu Iese, Karyn Morrissey, Predner Duvivier, Nigel Unwin.

**Formal analysis:** Leonor Guariguata, Gordon M. Hickey, Madhuvanti M. Murphy, Cornelia Guell, Nigel Unwin.

**Funding acquisition:** Madhuvanti M. Murphy, Cornelia Guell, Viliamu Iese, Karyn Morrissey, Predner Duvivier, Nigel Unwin.

**Investigation:** Leonor Guariguata, Madhuvanti M. Murphy, Cornelia Guell, Viliamu Iese, Karyn Morrissey, Predner Duvivier, Stina Herberg, Sashi Kiran, Nigel Unwin.

**Methodology:** Leonor Guariguata, Gordon M. Hickey, Madhuvanti M. Murphy, Cornelia Guell, Nigel Unwin.

**Project administration:** Viliamu Iese, Nigel Unwin.

**Resources:** Viliamu Iese.

**Supervision:** Gordon M. Hickey, Cornelia Guell, Nigel Unwin.

**Validation:** Stina Herberg, Sashi Kiran.

**Writing – original draft:** Leonor Guariguata, Gordon M. Hickey.

**Writing – review & editing:** Leonor Guariguata, Gordon M. Hickey, Madhuvanti M. Murphy, Cornelia Guell, Viliamu Iese, Karyn Morrissey, Predner Duvivier, Stina Herberg, Sashi Kiran, Nigel Unwin.

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
