## [Decision Letter · Decision Letter 0]

6 Jun 2023

PGPH-D-23-00287

Understanding the links between human health, ecosystem health, and food systems in Small Island Developing States using stakeholder-informed causal loop diagrams

Dear Dr. Unwin,

Thank you for submitting your manuscript to PLOS Global Public Health. I am pleased to invite you to submit a revision after considering the reviewers' suggestions and comments. Please respond to all the questions raised by the reviewers. Let me know if there is anything unclear.

We look forward to receiving your revised manuscript.

Kind regards,

Ying Zhang, PhD, MD

Academic Editor

Reviewers' comments:

Reviewer's Responses to Questions

**Comments to the Author**

1. Does this manuscript meet PLOS Global Public Health’s publication criteria? Is the manuscript technically sound, and do the data support the conclusions? The manuscript must describe methodologically and ethically rigorous research with conclusions that are appropriately drawn based on the data presented.

Reviewer #1: Yes

Reviewer #2: Yes

2. Has the statistical analysis been performed appropriately and rigorously?

Reviewer #1: N/A

Reviewer #2: N/A

3. Have the authors made all data underlying the findings in their manuscript fully available (please refer to the Data Availability Statement at the start of the manuscript PDF file)?

Reviewer #1: No

Reviewer #2: Yes

4. Is the manuscript presented in an intelligible fashion and written in standard English?

Reviewer #1: Yes

Reviewer #2: Yes

5. Review Comments to the Author

Reviewer #1: Introduction is quite long - paragraphs should be clearly structured and could be a little shorter. Writing style needs editing throughout and arguments are sometimes difficult to follow.

Line 55 - SIDS do not contribute that much to greenhouse gas emissions - this should be noted. I.e. SIDS food systems feel the impacts of climate change, but do not contribute to it significantly

Line66 onwards: “Adapting food systems to support food security and

66 sovereignty, reduce vulnerability to rapid environmental change and enhance productivity and

4

equity has been a major policy objective 67 in SIDS(9) with a recent commitment in the Caribbean,

68 for example, to reduce the regional food import bill by 25% by 2025” - there are a lot of concepts in this sentence which need to be more tightly linked together, e.g. environmental degradation is not directly linked with food imports

P84 “However, despite…: - I would start a new paragraph here. Also needs a clearer flow of argument/segue here as to why interventions targeting food security are related to climate change

Line 111: should this be pressing questions?

Line 121: “participatory complex systems thinking methods” needs a citation

Methods

Line 127 onwards - what was the time course of the project?

Would be good to include a table or more comprehensive list of the stakeholders involved

Results

Overall well done. Would be good, where possible, to see where the results you present came from e.g. which stakeholders or which countries said what. What is there all makes sense but more explicit links to the context of your data would help highlight the primacy and novelty of your findings.

Line 214: I don’t think the term ‘actors’ works for ‘four major actors’ as you use ‘actors’ elsewhere with a different meaning- could you use ‘factors’?

Line 240 “Political actors..” Should be a new paragraph

Line 249 onwards: this section contains a lot of really good points but could be more clearly structured to link the key issues e.g. agrochemical use, fossil food use, etc. to the key food production processes e.g. conventional farming, import of ultra-processed foods etc. At the moment it is all too jumbled together and the way it is presented the information loses its rigour.

Figure 2 - how is government commercial interests in unhealthy foods and agricultural practices dealt with?

Discussion

Line 406 - virtual group model

building sessions when face-to-face meetings were severely limited during the SARS-Cov

pandemic. - is this information in Methods?

Discussion is a little short - would be good to have a little more detail on examples of practical application of figures 1 and 2, and/or best first steps for governments and other stakeholders

Conclusion section is good, however I don’t see the immediate logic in the argument of this sentence: Global food systems are second only to energy production as a source of anthropogenic greenhouse gas emissions, and the ensuing climate change is itself making adequate food production more difficult.” - could this include an example of what element of SIDS food systems is CONTRIBUTING to climate change?

Reviewer #2: PGPH-D-23-00287

This manuscript makes an important contribution to the field, namely by identifying and describing the causal interrelationships among agricultural practices, food systems, climate and other forms of environmental change, and human health in Small Island Developing States (SIDS). Understanding these causes and effects is critical to build climate resilience and effective, community-driven adaptation strategies, as well as to improve food security and human health status. Even though this manuscript presents the results of a theoretical exercise rather than data from actual SIDS case studies, it highlights key sectors and actors that affect each other and need to be addressed to improve conditions on SIDS. This manuscript is very well-written and very clear.

Specific comments are listed below. Overall, they are minor with the exception of the observation that Figure 1 seems unnecessarily complex, and that Figure 2 contains “R”s that seem redundant with the clear relationships indicated with arrows.

Line-specific comments:

The long list of “interventions designed to adapt SIDS food systems” in lines 74-82 is important information, and it would be helpful to contextualize the results of their study if the authors explained these specific interventions more, have a few examples, and perhaps discussed briefly why they are insufficient (because the authors state that despite these interventions, “SIDS food systems remain highly vulnerable…”).

Lines 184-5: The authors list only one “indicator of food security (increasing dependence on food imports).” 1) This is actually an indicator of food insecurity, and 2) it should probably be acknowledged that there are also other indicators of food security.

Lines 209-211 and Figure 1: This figure is a bit complex, and the placements of “R”s and “B”s doesn’t always clearly indicate what relationship is being referred to. For example, why is “R8” listed where it is? And “R7” refers to two distinct relationships. (?) It also seems that there are both beneficial and harmful causal pathways, both in terms of the arrows used in the diagram and the “R”s, so could they be distinguished in this figure? Also, the term “ecosystem damaging unhealthy food” is misleading – rather than refer to “unhealthy food,” “agricultural practices” or “food system” is what is “ecosystem damaging.” I make further comments about Figure 1 in several comments below.

Line 215: Regarding “agroecological food interests” – this is the first time agroecology is mentioned, but this comment relates to the entire paper. It would be useful to clarify if by this term they mean any form of agricultural practice or food system that is not “conventional commercial.” Some people use the term “agroecology” in a general way to mean sustainable agricultural practices, but it can be defined in several ways, and an agroecological approach may involve any of numerous distinct farming methods. Best for the authors to clarify how they are using this term, because in this paper, “agroecology” is the sole alternative to “conventional commercial interests” and “ecosystem damaging unhealthy food” (see Figure 1). Many alternatives could be included.

Line 231: I would put a paragraph break here at “A climate supportive political…” In general, it would be easier to read and keep track of the explanations of the “R” feedbacks if shorter paragraphs are used.

Lines 241-247: This description from Fiji is an example of the relationship between politics and commerce and thus could be moved to line 231 (before the paragraph break suggested here).

Line 255: The subject word “They” needs to be defined for clarity.

Line 256: I recommend a paragraph break here as well, at the end of the line.

Lines 258 and Line 284 (and Figure 1): It seems a bit confusing why “R7” refers to two distinct feedback relationships. Can this be clarified?

Line 262: The “presence of pesticides” does not only “undermine healthy diets” and “damage the health of farmers” but also damages the health of food consumers, damages ecosystem health, contaminates water bodies, may destroy plants, and exposes various wildlife to potentially damaging toxic effects. Also “healthy diets” is a complex term that may be used even if conventionally grown foods (those that may have pesticide residues) are included. For example, one can define a “healthy diet” as full of fruits and vegetables, regardless of if they are grown conventionally or organically.

Line 266: “imported foods” are not necessarily of higher quality than “local agricultural products,” and may be more likely to contain conventionally-grown or processed food ingredients, pesticide residues, and other contaminants. This should be clarified.

Line 268: What specific sorts of “regulation” should be enacted to control “commercial producers” should be explained here.

Line 272: This is a good place to provide more detailed definition and explanation of what the authors mean by “agroecology” here. (See also comment above regarding line 215.)

Lines 279-281: It would be helpful to give a specific example of “agroecological food production” that illustrates how it is a beneficial alternative to “conventional monocrop food production practices in SIDS.”

Line 285: It would be helpful to define what the authors mean by “community-based principles.”

Line 286: Before saying that agroecology may “reinforce healthy diets,” first it should be stated that agroecology may produce better health outcomes driven by healthier diets, then say these diets may be reinforced.

Lines 290-302: This paragraph describes an important scenario whereby community cohesion leads to better food systems, but it is not a “given” that a community would be unified in its focus on local food productions, which approaches to food and farming would be chosen, and whether the entire community would support the same policies. If a “cohesive” approach is needed, it might be best to acknowledge that this “cohesion” probably would need to be cultivated, and how would that best be done?

(Line 303) and Figure 1: Why are B1 and B2 not explained?

Line 317 and Table 1: Clarify “how we define a community” – technically, isn’t this more about defining or identifying ideal roles of community members? And what is ideal may not necessarily be present in practice.

Line 319 and Table 1: For the “Role” of “Community members,” it is certainly desirable to have them be “co-creators and implementers of the intervention” but this isn’t necessarily always going to happen in all cases, right? Should this be clarified? Also, the second bullet point under “Description” of “Community members” is very specific – why is this intervention listed but not others?

Table 1 (continued): For the “Role” of the “Researcher,” the description makes it sound like “Community members” are not necessarily going to be creating interventions but that they may be imposed on them by “experts.” This wording should be clarified because it potentially contradicts the “Role” of “Community members.”

Table 1 (continued): For the “Description” of the “Funder,” it seems likely that funders should be responsive not only to researchers but also to community leaders and/or community-embedded partners, those actually implementing interventions.

Line 326: Indent?

Line 327: Shouldn’t “relationship” be plural?

Lines 330-1: Again, to clarify how agroecology is being conceived of here, it would be very helpful to give an example of how “implicit in that resilience is the adoption of agroecological principles.”

Figure 2: I’m not sure why the “R” feedback relationships are needed in Figure 2. The relationships are clearly described using text, and the authors don’t really describe these “R”s as distinct from the relationships indicated by arrows. This seems like different figure content than that in Figure 1.

Line 376: “policy and political” is awkward wording here. Seems like the authors could clarify that these are elected officials and decision-makers.

Line 399: Vulnerabilities of “children, adolescent girls, women and seniors” is not noted or described elsewhere, so it should not be brought up here without more description. It is an important point, but best not to casually bring it up in the conclusion without it being explained earlier. Also, it is not a given that attention to all the relationships in Figures 1 and 2, both positive and negative, will necessarily prioritize those most vulnerable. Focus on vulnerabilities, age and gender equity, etc. must be deliberate and cultivated through the use of specific equity lenses.

Lines 405-7: I am not an expert on the methods used here, so it would be helpful if the authors could explain how their approach “places control of the causal mapping firmly in the hands of the stakeholders.” How does that work? And how is it “flexible”?

In several places, semi-colon usage is incorrect: line 58, list in lines 74-82, lines 307-8, line 311

6. PLOS authors have the option to publish the peer review history of their article (what does this mean?). If published, this will include your full peer review and any attached files.

**Do you want your identity to be public for this peer review?** For information about this choice, including consent withdrawal, please see our Privacy Policy.

Reviewer #1: No

Reviewer #2: No

---

## [Editor Report · Decision Letter 1]

25 Aug 2023

Understanding the links between human health, ecosystem health, and food systems in Small Island Developing States using stakeholder-informed causal loop diagrams

PGPH-D-23-00287R1

Dear Professor Unwin,

We are pleased to inform you that your manuscript 'Understanding the links between human health, ecosystem health, and food systems in Small Island Developing States using stakeholder-informed causal loop diagrams' has been provisionally accepted for publication in PLOS Global Public Health.

Best regards,

Ying Zhang, PhD, MD

Academic Editor